# The Feasibility of Using N-Of-1 Trials to Investigate Deprescribing in Older Adults with Dementia: A Pilot Study

**DOI:** 10.3390/healthcare7040161

**Published:** 2019-12-12

**Authors:** Alexander J. Clough, Sarah N. Hilmer, Sharon L. Naismith, Danijela Gnjidic

**Affiliations:** 1Sydney Pharmacy School, Faculty of Medicine and Health, University of Sydney, Camperdown, NSW 2006, Australia; danijela.gnjidic@sydney.edu.au; 2Kolling Institute of Medical Research, Royal North Shore Hospital and Northern Clinical School, Faculty of Medicine and Health, University of Sydney, St Leonards, NSW 2064, Australia; sarah.hilmer@sydney.edu.au; 3Charles Perkins Centre, University of Sydney, Camperdown, NSW 2006, Australia; sharon.naismith@sydney.edu.au

**Keywords:** N-of-1 trials, deprescribing, dementia, statins, feasibility study

## Abstract

N-of-1 trials may provide insights into the impact of deprescribing medications in populations where evidence is currently lacking, such as the effect of statins on cognition in people with dementia. For this pilot, N-of-1, double-blinded, deprescribing trial, adults over 80 years of age with dementia taking statins for at least 6-months were recruited from a hospital’s geriatric medicine outpatient clinic in Sydney, Australia. Participants discontinued and restarted statins over the study period. At enrolment, the hospital pharmacy—using a random number generator, randomised recruited participants to their usual statin or placebo regimen, with assessment and switching of treatment every 5 weeks from baseline (0-weeks) until Visit 4 (15-weeks). Primary outcome was measured using the rate of change in Alzheimer’s Disease Assessment Score-Cognitive Subscale (ADAS-CoG). Over 6-months, 81 participants were screened, 14 were eligible, and four were randomised. One participant (female, 88 years) completed all four assessments with no major harms reported. Cognitive impairment, as measured by ADAS-CoG score, was similar on placebo (15.5/70) compared to statin (15/70). This study suggests there are significant challenges in performing N-of-1 trials and recruiting people with dementia into deprescribing trials from outpatient settings.

## 1. Introduction

Deprescribing is the supervised discontinuation of inappropriate medication(s) [1]. At present, the process of implementing deprescribing into clinical practice and performing clinical trials is challenging, especially in vulnerable patient groups, such as older adults with dementia [1]. Older adults with dementia are particularly prone to suffering adverse drug events (ADEs) and subsequently, poorer health outcomes compared to older adults without dementia [2,3]. One clinical area that continues to be debated is the effect of statins on cognition, especially as there is uncertainty regarding their efficacy for the primary prevention of cardiovascular disease in people over 80 years of age [4]. A recent meta-analysis of 28 trials highlighted the lack of direct benefits among adults older than 75 years [5]. Furthermore, evidence from an open-label study suggests that cognition may improve upon discontinuation of statins and worsen upon rechallenging [6].

However, to generate evidence on the benefits and harms of deprescribing medications, clinical trials need to be conducted [7]. Currently, there is a lack of consensus on the most applicable trial methodology to inform the efficacy and safety of deprescribing medications [7]. N-of-1 trials can provide information in patient populations where the effects of medications on particular disease states is poorly understood [8]. For example, N-of-1 trials have been previously used to determine the optimum treatment for people with chronic diseases, such as schizophrenia, and can overcome the heterogeneity of treatment effects in these populations [9,10,11]. Inclusion of people with chronic diseases, who are generally excluded in clinical trials, appears to be safe, indicating the potential of recruitment involving older adults with cognitive impairment in N-of-1 trials [10,11]. N-of-1 trials have also been used in studying the side effects of statins, specifically, statin-related myalgia in older adults 65-years-old and above, indicating their potential for investigating the relationship between statins and other non-specific outcomes such as cognitive impairment [12]. Furthermore, deprescribing N-of-1 trials are safe and can generate strong patient-specific evidence on the benefits and harms of discontinuing medications, potentially informing and facilitating the conduct of subsequent, robust, clinical deprescribing trials [13].

The aim of this pilot study was to examine the feasibility of conducting an N-of-1, deprescribing, trial in older adults with dementia, and to generate pilot data on the short-term impact of deprescribing statins on cognition in older adults with dementia.

## 2. Materials and Methods

Ethics approval was granted by Northern Sydney Local Health District Human Research Ethics Committee (HREC/16/HAWKE/286) and registered on the Australian New Zealand Clinical Trials Registry (ACTRN12617000214336). This study is reported per the Consolidated Standards of Reporting Trials (CONSORT) Extension for N-of-1 Trials (CENT) checklist [14].

A pilot, N-of-1, double-blinded, deprescribing trial was conducted with participants randomised to Treatment Arm 1 (placebo-statin-placebo) or Treatment Arm 2 (statin-placebo-statin) for a period of 15 weeks. Each intervention period was 5 weeks in duration with no washout (Figure 1). Single person, or N-of-1 trials, use a randomised or balanced study design characterised by periodic switching from active treatment to placebo over time.

Participants were recruited from a geriatric outpatient clinic at Royal North Shore Hospital (RNSH), Sydney, NSW, Australia. Participants were approached for recruitment by study physicians and formally invited to participate by the researcher, if deemed eligible and suitable for enrolment (AJC). Informed written consent was obtained by the participant if deemed competent, or from the person responsible, if required.

Participants were included if they: were aged 80 years or more; had a Mini-Mental Examination (MMSE) [15] score between 10 and 23 (indicating mild to moderate cognitive impairment); had a clinical diagnosis of dementia; were taking a statin for at least 6 months; and, were willing to give written and verbal consent and to participate in the study, and/or give verbal assent with their person responsible gave written and oral consent. Participants were excluded if they: were too unwell to participate or in the terminal phase of an illness; had an active cancer diagnosis; taking a combination statin medication; and, were unable to provide informed consent, and/or person responsible did not consent for participant to participate. The study was advertised on the Dementia Australia website and with pamphlets in the clinic.

The primary outcomes included:Feasibility of using the N-of-1 method for this clinical deprescribing trial, assessed by analysing the actual recruitment numbers compared with screening and eligibility numbers.Change in Alzheimer’s Disease Assessment Scale-Cognitive Subscale (ADAS-CoG) score of each participant [16]. The ADAS-CoG is a comprehensive psychometric instrument that evaluates: memory, attention, reasoning, language, orientation, and praxis. Scores range from 0 to 70 with a higher score indicating an increased degree of cognitive impairment. A mean difference of at least 4 points between intervention periods was to be considered as clinically significant for this study. This measure was chosen as it has been used in previous Alzheimer’s disease (AD) statin trials [17]. Furthermore, the test is used to inform on the efficacy and cost effectiveness of Pharmaceutical Benefits Scheme (PBS) subsidised medicines for the treatment of AD, and there are correlations between improvement in ADAS-CoG score and clinically meaningful measures in people with moderate dementia [18].

Secondary outcomes included patient-relevant, carer-relevant, and physical measures. These were assessed using:Self-reported Quality of Life in Alzheimer’s Disease (QoL-AD), as per previous clinical trials in dementia, and as also recommended by the Dementia Collaborative Research Centres, Australia [19]. It is a brief, 13-item measure recording the participant’s quality of life and is completed by the participant as an interview, and by their caregiver as a questionnaire. A higher score indicates greater quality of life [20].The Short Performance Physical Battery (SPPB); a series of physical tests mimicking daily activities. It is scored from 0 to 12 with a higher score indicating greater physical performance [21].The John Hodges Cambridge Behavioural Inventory—Revised for the carer (CBI-R); a carer-completed questionnaire on observed cognitive and behavioural disturbances that are used to determine carer-related outcomes. It has 45 questions, scored on a scale of 0 to 4 with larger scores indicating more frequent problems [22].

In addition to outcome data, socio-demographic data including age, gender, marital status, living status, primary carer, and ethnicity were collected at baseline. Medications taken by the participants, including statin and dosage, were recorded, and comorbidities were determined using the Charlson Comorbidity Index at baseline [23]. A single investigator (AJC) performed all assessments at the participants’ homes.

Individual participants were randomised by the hospital pharmacy to placebo-statin-placebo (Treatment Arm 1) or statin-placebo-statin (Treatment Arm 2). For example, if a participant was randomised to Treatment Arm 1, they were given a placebo for a period of 5 weeks and then crossed over to receive their standard statin treatment for a further 5 weeks. Following this, participants were given a placebo for 5 weeks with conclusion at the end of this intervention period, after 15 weeks of total trial participation. At trial conclusion, participants resumed their original statin medication (Figure 1). There was no washout. Assessments of primary and secondary outcomes were performed every 5 weeks at each visit from baseline (Visit 1) to conclusion (Visit 4), resulting in a total of four assessments.

Placebo and statin medications were over-encapsulated to ensure treatment allocation and maintain blinding of participants, carers, and researchers. Statin or placebo capsules were prepared by a pharmaceutical company, Pharmaceutical Packaging Professional in Melbourne, Australia, and delivered to, and dispensed by, the hospital’s clinical trials pharmacy. The dose for each participant was set by that used by the participant at recruitment. The statin medication provided matched participants’ usual statin medication in dose and route of administration at the time of recruitment. Lastly, to ensure adherence to the treatment allocation, pill count and returned capsule count method was used to assess compliance.

Participants were randomised to each treatment arm using a random number generator after enrolment, with randomisation conducted by the hospital pharmacy blinded to the research team. Each treatment arm consisted of 3, randomised, 5-week intervention periods, comprising 15 weeks of trial duration; in line with other studies investigating the short-term effects of statins on cognition [12,24].

Sequence allocation was concealed by labelling each bottle of medicines with information on placebo and on statin medication. Furthermore, statin medication and placebo tablets were over-encapsulated to ensure blinding of researchers and participants.

At trial conclusion, researchers were informed of sequence allocation by the hospital pharmacist. Allocation and results were presented to participants and their general practitioners (GPs) after trial conclusion, so that a decision could be reached on whether a participant should continue, adjust, or discontinue their statin medication. Follow-up data was collected from the GP on any decision made one month after presentation of results.

ADEs and adverse drug withdrawal events (ADWEs) were documented at all time-points and compared between the groups.

As this was a pilot, feasibility study, no formal power calculations were performed. However, as per previous studies, and the protocol, 30 participants were planned for recruitment [17].

All data were transferred from data collection sheets into IBM Statistical Package for Social Sciences (SPSS Version 24) Statistics. Descriptive analysis was performed to summarise the study population characteristics using means and standard deviation (SD). Data analysis was conducted within each N-of-1 trial and results combined across them following the intention-to-treat principle of analyses. The difference in mean ADAS-CoG scores between statin medication and placebo was the pre-specified primary outcome. The t-test approach was used to obtain the treatment effect point estimates and the 95% confidence intervals to compare the mean ADAS-CoG scores. Descriptive analysis was used to summarise secondary outcomes as appropriate.

## 3. Results

Over the course of 6 months (January to June 2017), 81 people were screened, 14 were eligible, four participants were enrolled, and one participant completed the study from January to June 2017 (Figure 2). Baseline characteristics are shown in Table 1. The mean age was 85.1 years, majority were female (75%), two participants were taking atorvastatin (one on 20mg and one on 10mg) at enrolment, with a mean MMSE score of 17.5/30. One participant completed all study visits.

Randomisation schemes, and results of the N-of-1 trial are presented in Table 2. The mean cognition scored, as assessed by the ADAS-CoG when participant was 15.5/70 on placebo and 15/70 on statin. Regarding secondary outcomes, mean quality of life score was 37/52 and 34/52; mean physical function score was 7/12 and 6.5/12; and, mean carer-related score was 21/180 and 25.5/180 when the participant was on statin and on placebo, respectively. The participant complained of mild foot pain at the second visit, remaining for the duration of the trial. The participant had 100% adherence to the study protocol.

The participant’s GP was contacted at the conclusion of the trial with the results, and a short explanation, of their patient’s N-of-1 trial. A reply was requested in order to determine if the participant’s statin regimen had been altered, however, there was no response from the GP.

There were several recruitment barriers identified and reported by the recruiting physicians in this study. These include, treating physicians being unwilling to deprescribe statins, time constraints in outpatient clinics, and participant stress after a clinical diagnosis of dementia. The physicians reported it was not feasible to recruit the target number of participants in the geriatric outpatient setting used in this trial.

## 4. Discussion

In this pilot N-of-1 deprescribing trial, we assessed the feasibility of using the N-of-1 trial method to investigate the short-term impact of discontinuing and rechallenging statins on cognition in older adults with dementia. The results of this study suggest that there are major barriers to recruiting older adults with dementia into deprescribing trials from outpatient settings, with the target of 30 participants not met. While the study protocol was feasible for the one participant enrolled, the generalisability of this finding is not clear. Moreover, this study was not powered to investigate differences in cognition, nor in global functioning and functional status with the discontinuation and rechallenging of statins.

To our knowledge, this was the first study to investigate the feasibility of using the N-of-1 deprescribing trial method for determining the impact of statins upon cognitive impairment in adults 80-years-old and over with dementia in the outpatient setting. Most studies, to date, have indicated that statins may have a protective benefit on cognition. However, the evidence is inconclusive, generated primarily from observational studies, not been deprescribing studies, and previous randomised controlled trials have lacked quality [17,25,26,27,28,29]. Despite this however, a warning label concerning the potential impacts of statins was added by the US Food and Drugs Administration (FDA), citing potentially reversible cognitive side-effects [25]. This has been supported through a statin deprescribing study where participants discontinued statins for 6-weeks and were then rechallenged for 6-weeks [6]. It was found that when statins were ceased, cognition improved, and when statins were restarted, cognition declined [6]. However, this study was open-label and a Cochrane review found no randomised controlled deprescribing studies to support this evidence [30].

Previous application of the N-of-1 methodology to statins and myalgia demonstrated that in eight people who had previously reported myalgia, there was no significant difference in myalgia scores for any participants between statin and placebo, and five resumed active statins upon trial conclusion [12]. Other previous, N-of-1 deprescribing trials have provided patient-specific evidence to participants, resulting in the discontinuation on non-beneficial treatments, informing future prescribing decisions [13]. This strong patient-specific evidence informed our study, whereby the discontinuation and rechallenge of statins over time could inform on the feasibility of using N-of-1 trials to investigate whether statins impact cognition in cognitively impaired older adults.

Our study was hampered by significant challenges in recruiting potential participants. Firstly, the recruitment setting of a single geriatric outpatient clinic may not be the most feasible recruitment site. Physicians, responsible for initially approaching potential participants and outlining the trial aims and methodology, commonly specified that it was not always feasible to conduct the recruitment process during the clinic appointment. At the end of the screening process, some physicians expressed that patients were primarily visiting the clinic after being referred by their usual GP to establish a formal dementia diagnosis. Indeed, this was an additional barrier to recruiting participants. Physicians indicated that their patients were experiencing a lot of stress due to a new dementia diagnosis, and they felt it was unsuitable to recruit these vulnerable people into a clinical trial at the point of establishing a newly-diagnosed dementia. These opinions are consistent with concerns regarding recruitment rates of older adults with dementia in clinical trials, which is consistently low with only 26% of people with Alzheimer’s disease eligible for most trials [31,32].

Secondly, physicians involved in the recruitment process commonly expressed the time pressures during clinic appointments, especially the time taken to communicate the process of establishing a formal diagnosis. They stated there is not enough time in the visit to outline all the features of a clinical trial. Finally, some physicians did not feel comfortable deprescribing statins in this cohort due to the previously outlined conflicting evidence impact on cognition [29]. The hesitance of physicians to deprescribe has been reflected in other literature on the issue, with intrinsic barriers commonly identified [33]. However, older adults have previously indicated a willingness to have their statins deprescribed, and many geriatricians have expressed that limited life expectancy and cognitive impairment are strong drivers of deprescribing medications in clinical practice [34,35].

There are a few strengths regarding this study though. Firstly, this pilot, N-of-1 trial was double-blinded with a single investigator performing all assessments, minimising information bias. Medication inventory was performed by checking all medications at the participant’s home. The use of the N-of-1 trial method allowed for the comparison of treatment effects within a single participant and was tolerable for the participant, with all assessments performed at the participant’s home in the presence of their carer.

However, there are important limitations to our study, mainly related to recruitment challenges, which limited the ability to evaluate the feasibility of the study protocol to one participant. The recruitment rate was low, with only 14 of 81 (17.3%) screened patients being eligible to take part in the study and one completing all assessments. Furthermore, several recruitment barriers were identified by physicians in this study, including time constraints, which should be used to inform recruitment strategies from future studies. However, it should be noted that recruitment was only performed in one geriatric outpatient clinic, and different barriers may be identified in other studies. The relatively short time of interventions may have limited the assessment of the effects of discontinuing and restarting statins due to the time to effect, however we wished to investigate short-term effects of statin discontinuation and did not wish to burden participants with a long trial duration.

Future studies should explore recruiting participants from community settings where there may be fewer stressors for the participant and recruiting doctors or use video consent to overcome time pressures. Additionally, future studies could explore the possibility of developing a clinical deprescribing trial framework to guide future trials. On the deprescribing of statins, more high-quality evidence is needed to guide physicians on appropriate treatment for older adults, which may be facilitated through N-of-1 trials, or other robust trial methods [36].

## 5. Conclusions

In conclusion, this pilot N-of-1 deprescribing trial demonstrates that it is not feasible to recruit people with dementia to assess the impact of statins on cognition in older adults with dementia from a geriatric medicine outpatient clinic. The study identified major barriers to recruiting older adults with dementia into deprescribing trials from outpatient settings including: treating physicians being unwilling to deprescribe statins; time constraints in the clinic; and, participant stress after clinical diagnosis of dementia. These barriers should be addressed in future deprescribing trials enrolling people with dementia, by potentially recruiting from other settings.

## Figures and Tables

**Figure 1 healthcare-07-00161-f001:**
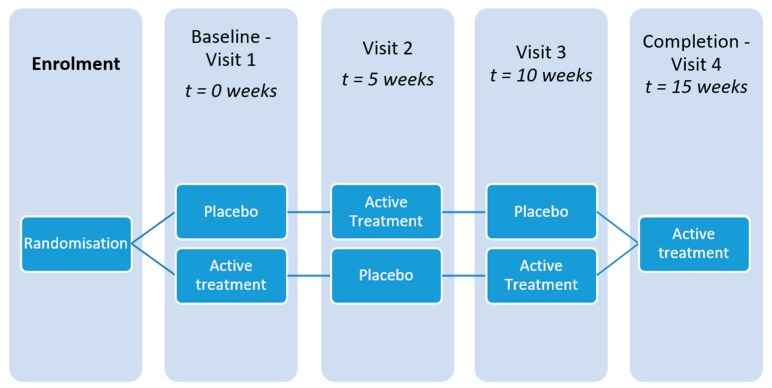
N-of-1 trial study procedure. Active treatment = normal statin dose and administration.

**Figure 2 healthcare-07-00161-f002:**
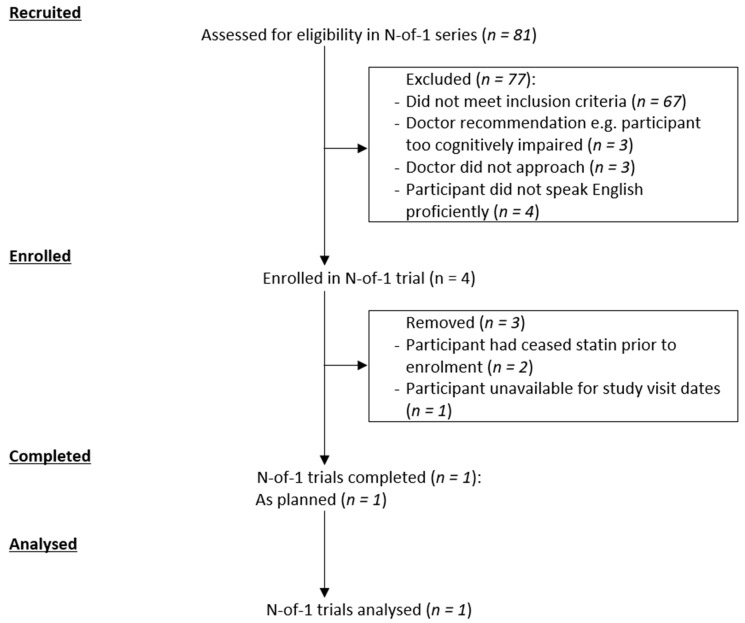
Participant flowchart.

**Table 1 healthcare-07-00161-t001:** Baseline characteristics of enrolled participants.

Characteristic	All Participants
Mean Age (Years)	85.1 ± 3.4
Females, *n* (%)	3 (75)
Statin, type (daily dose) (*n*)	Atorvastatin (20 mg (1), 10 mg (1)) Rosuvastatin (5 mg (1)) ^1^
Mean Mini Mental State Examination Score	17.5 ± 2.9

^1^ For one participant, statin type and dose were not recorded (participant unavailable for study visit dates).

**Table 2 healthcare-07-00161-t002:** Cognition and global functioning score with discontinuation and rechallenged of statins in an N-of-1 trial in one participant who completed study.

Intervention	Visit Number (time-point)	ADAS-CoG ^1^ (/70)	SPPB ^2^ (/12)	QoL-AD ^3^ (/52)	CBI-R ^4^ (/180)
Active ^5^	One (baseline)	15	9	40	20
Placebo	Two (5-weeks)	12	7	39	19
Active	Three (10-weeks)	15	5	34	22
Placebo	Four (15-weeks)	19	6	29	32
**Overall change (mean active − mean placebo)**		**−0.5**	**0.5**	**3**	**−4.5**

^1^ ADAS-CoG = Alzheimer’s Disease Assessment Score − Cognitive Subscale. ^2^ SPPB = Short Performance Physical Battery. ^3^ QoL-AD = Quality of Life in Alzheimer’s Disease. ^4^ CBI-R = Cambridge Behavioural Index − Revises for the carer. ^5^ Active = normal statin dose and administration.

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
