# Peer review of "The Feasibility of Using N-Of-1 Trials to Investigate Deprescribing in Older Adults with Dementia: A Pilot Study"

_healthcare, 2019, doi:10.3390/healthcare7040161_

Round 1

Reviewer 1 Report

This was an interesting concept as a study, however far too complicated for what is needed in practice. Personally did not see the point of stopping a medication which has no evidence-based value on this population group (only if someone had familial hypercholesterolaemia) only to restart it. Seems non-sensical. Especially as in younger people it takes 77 hours for atorvastatin to be eliminated from the body and one would expect that to take longer in older people.

The evidence we need is what happens to older people when they stop statins. Anecdotal evidence suggests  that mobility improves because muscle pain decreases, people sleep better as the effect on dreaming ceases and cognitive function increases. Not sure the first two were measured. 

Having no washout period meant that someone may have jut been recovering from the side effects of a statin, when it was restarted. If a medication has no value in this age group them stop it. Monitor it over time. For a statin over at least a month. Do not see the point of a trial for this; makes things far too complicated in a vulnerable group who should not be stopping things just to restart them. If there is no intrinsic value, withdraw with patient, GP and carer consent. If needed you could check prescribing databases where the population data exists for people over 8- stopping statins and followup outcomes that way.

N-1 process will always be hampered by prescribers' who do not want to interfere in another prescriber's prescribing. Ask the patient, their carer and the GP at a time that is not stressful to patient, carer and secondary prescriber.

This was a well written paper and a well executed study (if over complicated for this patient population group). For people with dementia we need to simplify medication regimens, not over-complicate them. 

Worth publishing just to alert people that this may not be a suitable method for this population group

 Reviewer 2 Report

Authors performed an investigation of switching placebo and active treatment in patients with dementia for visits separated by five weeks. This feasibility study of N/1 trails for assessing the older adults with dementia provided a reasonable design of experiments. These N/1 gave insights into the impact of deprescribing medications in populations where little evidence is available the effect of statins on cognition in people with dementia. Experiment consisted of double-blinded, deprescribing trial for adults with 80+ years of age with dementia. These patients were recruited from a hospital’s geriatric medicine outpatient clinic in Sydney, Australia where they were taking statins for at least 6-months. The patients were made to discontinue and restart the statins over this study period. Participants were observed over baseline (0-weeks) to visit-4 (15 weeks) where the gap of 5 weeks ensure enough time of change. For enrolling the patients, hospital employed a random number generator-based scheme which made the process unbiased.  Participants were made to switch between usual statin and placebo regimen.

Authors used rate of change in Alzheimer’s Disease Assessment Score-Cognitive Subscale (ADAS-CoG) for measuring the primary outcome of this investigation. The motivation of choosing this scale is not present in this manuscript. Explicit mention of objective reasons for choosing this scale can help increase credibility of shown results. In this 6-month study, out of 81 screened participants, 14 were eligible, and four were randomized. One female patient aged 88 years completed all four assessments with no major harms reported. This case measured same ADAS-CoG score for cognitive impairments, as measured by placebo (15.5/70) and statin (15/70).

Overall, the impact of proposed study suggests significant challenges in performing N-of-1 trials and recruiting patients with dementia into deprescribing trials. These experimental trials demonstrate that it is not feasible to recruit people with dementia to assess the impact of statins on cognition in older patients. The reason being These patients with dementia are from a geriatric medicine outpatient clinic. This research found out main bottlenecks in conducting such trials including (i) physicians are sometimes unwilling to deprescribe statins; (ii) time is limited in clinic for fair usage for each patient; and, (iii) patients can get stressed diagnosis of dementia. Knowing these bottlenecks help to design techniques to tackle them in near future.

Some spelling mistake “randomised” -> “randomized” can be found at several instances. Authors are advised to do a round of proof-reading, grammer and spell-check that will improve the readability of the paper.

  • In line with the journal’s policies, we have consistently used UK English, but we have conducted a full grammar review.

References:

Joy, T. R., A. Monjed, G. Y. Zou, R. A. Hegele, C. G. McDonald, and J. L. Mahon. 2014. 'N-of-1 (single-patient) trials for statin-related myalgia', Ann Intern Med, 160: 301-10.

McGuinness, B., D. Craig, R. Bullock, R. Malouf, and P. Passmore. 2014. 'Statins for the treatment of dementia', Cochrane Database Syst Rev: Cd007514.

Padala, K. P., P. R. Padala, D. P. McNeilly, J. A. Geske, D. H. Sullivan, and J. F. Potter. 2012. 'The effect of HMG-CoA reductase inhibitors on cognition in patients with Alzheimer's dementia: a prospective withdrawal and rechallenge pilot study', Am J Geriatr Pharmacother, 10: 296-302.

van der Ploeg, M. A., C. Floriani, W. P. Achterberg, J. M. K. Bogaerts, J. Gussekloo, S. P. Mooijaart, S. Streit, R. K. E. Poortvliet, and Y. M. Drewes. 2019. 'Recommendations for (Discontinuation of) Statin Treatment in Older Adults: Review of Guidelines', J Am Geriatr Soc.

Verhey, F. R., P. Houx, N. Van Lang, F. Huppert, G. Stoppe, J. Saerens, P. Bohm, L. De Vreese, A. Nordlund, P. P. DeDeyn, M. Neri, J. Pena-Casanova, A. Wallin, E. Bollen, H. Middelkoop, M. C. Nargeot, M. Puel, U. M. Fleischmann, and J. Jolles. 2004. 'Cross-national comparison and validation of the Alzheimer's Disease Assessment Scale: results from the European Harmonization Project for Instruments in Dementia (EURO-HARPID)', Int J Geriatr Psychiatry, 19: 41-50.